# Assessment of TNF-α, IL-12/23, and IL-17 in Psoriasis: Only TNF-α Reflects Clinical Response After 12 Weeks of Biologic Treatment

**DOI:** 10.3390/cimb47050368

**Published:** 2025-05-16

**Authors:** Alessandra-Mădălina Matei-Man, Ștefan Cristian Vesa, Alexandra Dana Pușcaș, Meda Sandra Orăsan, Bianca Homorozeanu, Elisabeta Candrea, Teodora Mocan

**Affiliations:** 1Dermatology Department, Faculty of Medicine, “Iuliu Hațieganu” University of Medicine and Pharmacy, 400126 Cluj-Napoca, Romania; madalina.man@outlook.com (A.-M.M.-M.); elisabeta.candrea@umfcluj.ro (E.C.); 2Department of Pharmacology, Toxicology and Clinical Pharmacology, “Iuliu Hatieganu” University of Medicine and Pharmacy, 400337 Cluj-Napoca, Romania; 3Independent Researcher, 1021 Budapest, Hungary; dr.alexandradana@gmail.com; 4Physiopathology Department, Faculty of Medicine, “Iuliu Hatieganu” University of Medicine and Pharmacy, 400126 Cluj-Napoca, Romania; orasan.meda@umfcluj.ro; 5Department of Radiation Oncology, “Iuliu Hatieganu” University of Medicine and Pharmacy, 400012 Cluj-Napoca, Romania; homorozeanubianca@gmail.com; 6Physiology Department, Faculty of Medicine, “Iuliu Hatieganu” University of Medicine and Pharmacy, 400126 Cluj-Napoca, Romania

**Keywords:** psoriasis, biomarkers, biologic therapy, TNF-α

## Abstract

Background: Tumor necrosis factor-alpha (TNF-α), IL-12/23, IL-17A, and IL-17F are key proinflammatory cytokines involved in the pathogenesis of psoriasis. Biologic therapies targeting these interleukins have demonstrated clinical efficacy. However, the exact relationship between their serum levels and clinical response remains unclear. The aims of this study are to assess changes in cytokine levels (TNF-α, IL-12/23, IL-17A, IL-17F) after 12 weeks of biologic treatment in psoriasis to test if there is any correlation between their serum level and PASI (Psoriasis Area Severity Index) and DLQI (Dermatology Life Quality Index) scores before or after treatment and to check the influence of clinical and lifestyle factors on these levels. Methods: In this prospective study, 36 patients with moderate-to-severe plaque psoriasis receiving anti-TNF-α, anti-IL-17, or anti-IL-23 therapy were assessed at baseline and after 12 weeks. The serum levels of these cytokines were measured using the ELISA technique. Clinical response was evaluated using PASI and DLQI scores. Spearman correlation analysis was used to assess the relationship between interleukins’ serum levels and these scores. Results: A significant decrease in TNF-α levels and DLQI and PASI scores was observed after 12 weeks across all treatments. A moderate positive correlation (r = 0.391, *p* = 0.018) was found between serum TNF-α levels and PASI scores at week 12. Conclusions: The serum levels of TNF-α are significantly correlated with PASI scores following 12 weeks of biologic therapy, supporting their potential role as a biomarker for monitoring treatment efficacy in psoriasis.

## 1. Introduction

In the current times, clinical diagnosis and follow-up is the conventional approach for psoriasis. Advances in understanding psoriasis pathogenesis have led to the development of biologics, which are molecules directed against key cytokines incriminated in psoriasis progression (TNF-α blockers, IL-12/23p40 inhibitors, IL-23 inhibitors, IL-17A inhibitors, IL-17A/F inhibitor, and IL-17R antagonist). These findings point out the potential role of these cytokines as biomarkers for guiding treatment choices, because they mirror the critical disease processes [1].

A biomarker is an objective quantifiable trait reflecting the normal physiological processes, disease progression, or the response to a certain therapy. Although, biomarkers can be divided into the following three groups: those that are associated with the disease’s natural course, those that show the impact of an intervention based on a drug’s pharmacological mechanism, and biomarkers that serve as surrogate endpoints for clinical outcomes. In psoriasis, biomarkers have been investigated for the diagnostics and quantification of disease severity but also for evaluating conventional or biologic treatment response and for identifying the risk of related comorbidities, like psoriatic arthritis, cardiovascular disease, liver fibrosis, or kidney disorders [2].

Skin biopsy is an invasive and not always feasible tool for routine monitoring in psoriasis patients. Contrarily, blood-based biomarker assays are minimally invasive and allow repeated measurements to trace disease activity. The sandwich ELISA technique is a widely available, specific, and sensitive method for biomarker quantification [1].

To identify and validate suitable biomarkers for effectiveness and prognosis in the treatment of psoriasis, one must conduct accurate studies to find correlations with clinical outcomes, such as disease severity scales and treatment response. Different studies have shown an increased expression of various proinflammatory cytokines in both the skin and serum of patients with psoriasis [3].

The rationale for investigating these biomarkers stems from studies demonstrating that serum levels of TNF-α, IL-17A, IL-17F, and IL-12/23 play a critical role in the development of psoriasis clinical manifestations and response to treatment.

However, results are often conflicting, because they are conducted on a reduced number of participants, or the observations are limited to a short period of time. To rule out a biomarker sensitivity, specificity, and clinical applicability, there is an imperative need for large-scale validation. The assimilation of biomarker analysis into everyday clinical practice could improve disease monitoring and reform therapeutic strategies, ultimately enhancing patient outcomes [1].

To the best of our knowledge, there is a lack of studies and data examining the longitudinal effect of biological treatment on cytokine levels within the same group of patients.

This research aimed to achieve the following:Determine whether the serum levels of TNF-α, IL-17A, IL-17F, and IL-12/23p40 decrease after biological treatment;Check if the PASI score is correlated with the serum levels of the assessed cytokines before or after 12 weeks of biological treatment;Check if the treatment class, disease duration, obesity, alcohol consumption, smoking status, or gender influences the serum levels of the assessed cytokines before or after 12 weeks of biological treatment.

## 2. Materials and Methods

### 2.1. Design and Participants

This study is a longitudinal analysis of biologically naïve patients with psoriasis vulgaris. We collected serum samples form patients visiting our clinic between 2023 and 2024. Inclusion criteria required patients to be over 18 years old, have a confirmed diagnosis of psoriasis (via histopathological examination), be biologically naïve, and require the initiation of biological therapy due to the ineffectiveness of a previous conventional systemic or topical treatments. Exclusion criteria were age below 18 years old, psoriasis forms that responded well to conventional therapy, or bio-experienced patients. A total of 36 patients were enrolled, all of whom started biological therapy with anti-TNF-α agents (Adalimumab, Infliximab, Certolizumab pegol, Etanercept), anti-IL-17 agents (Ixekizumab, Secukinumab), and anti-IL23 monoclonal antibodies (Tildrakizumab, Risankizumab). The selection criteria for biologics in patients was guided by several clinical and individual factors, such as moderate-to-severe cases of psoriasis vulgaris, defined as PASI score greater than 10, DLQI above 10, documented history of incomplete response, intolerance or contraindications to conventional systemic therapies (methotrexate, cyclosporine), the presence of comorbidities like psoriatic arthritis, inflammatory bowel disease or cardiac failure, safety considerations, including infection risk and malignancy history, and lastly, patient-related aspects, such as treatment preferences and reproductive plans. A follow-up visit occurred 12 weeks after treatment initiation. The 12-week interval was chosen based on standard clinical assessment intervals and biologic treatment response timelines.

Psoriasis severity was assessed during both visits using PASI and DLQI scores by the treating dermatologist (Orăsan Remus-Ioan), as well as general examination.

The DLQI score is a questionnaire created to assess how dermatological diseases impact a patient’s quality of life. It contains 10 questions, which cover symptoms, feelings, and impact of the disease on daily activities, leisure, and working or learning capacity. The patient will give a rating from 0 to 3 (no impact to high impact), resulting in a total score of a maximum of 30. A score of 0–1 states for no impact on quality of life, 2–5 reflects a small impact, 6–10 moderate effect, 11–20 a significant effect, while 21–30 score is linked to severe impairment on quality of life.

The PASI score is mainly used in research to assess the severity of psoriasis considering the characteristics of the plaques (erythema, induration, thickness, and scaling) and the body surface affected in four areas (head, upper limbs, trunk and lower limbs). One can have a maximum score of 72. A score of 5 to 10 is considered moderate disease, while a score above 10 states severe disease. The PASI score can also be a useful tool to assess the effectiveness of different treatments suitable in psoriasis.

During both visits, blood samples were collected in 5 mL sodium citrate tubes, then they were left at room temperature for 30 min and centrifugated at 1000 rpm for 10 min. Subsequently, the serum layer was removed and stored at −80 °C until analysis.

Total serum cytokine levels (TNF-α, IL-17A, IL-17F, and IL-12/23p40), including inactive forms, were measured using BioLegend^®^, San Diego, CA, USA. Human TNF-α, IL-17A, IL-17F, and IL-12/23 (p40) ELISA Kits, following manufacturer’s protocol. These kits use Sandwich ELISA technique, with pre-coated microplates containing specific antibodies for each cytokine. Standards and samples were added to the wells, and the specific interleukins bounded to the immobilized capture antibody. Next, a biotinylated human detection antibody was added, producing a blue color in proportion to the concentration of the interleukin assessed. This step changed the reaction color from blue to yellow. The absorbance in wells was then read at 450 nm using a microplate reader.

This study was conducted in accordance with the Declaration of Helsinki and received approval from the Ethics Committee of the “Iuliu Hațieganu” University of Medicine and Pharmacy (approval number 145/4 July 2023). All patients provided written informed consent before participation.

### 2.2. Statistical Analysis

Data processing was performed with MedCalc^®^ Statistical Software version 23.1.6 (MedCalc Software Ltd., Ostend, Belgium; https://www.medcalc.org, accessed on 23 April 2025). We analyzed the variation in measured parameters before and after 12 weeks of treatment. We used Wilcoxon signed rank test to compare the values of the assessed proinflammatory cytokines and the severity scores at the start of the biological therapy and after 12 weeks of treatment. The ANOVA test for repeated measurements was also used for comparing the effectiveness between biological agents, disease duration, obesity, alcohol consumption, smoking, and gender in reducing the level of interleukins. The correlations between cytokine values and the PASI and DLQI scores were tested using Spearman’s rho correlation coefficient. The limit of statistical significance was considered *p* < 0.05.

## 3. Results

We enrolled 36 patients in the study, whose general characteristics are summarized in Table 1. For the purpose of this study, short disease duration (SDD) was defined as a time period of less than 2 years from disease onset to the initiation of biologic therapy, while long disease duration (LDD) was defined as more than 2 years.

In total, 15 (41.7%) patients received anti-IL-23 treatment, 13 (36.1%) anti-TNF-α, and 8 (22.2%) anti-IL-17 biological treatment. At baseline, the group of patients who received anti-TNF-α biologic therapy had a median PASI score of 25.61 (18; 22.2) and a median DLQI of 26.84 (18; 25). Patients treated with anti-IL17-A therapy had a median PASI score of 19.27 (13.25; 23.5) and a median DLQI of 18.87 (12.75; 24.5), while those who received anti-IL-23 therapy had a median PASI score of 19.13 (14.4; 22.35) and a median DLQI of 19.3 (15; 23.5). A total of 17 patients had at least one comorbidity, with some patients presenting multiple comorbid conditions. Of the seven patients with hypertension, two were treated with anti-TNF-α agents, and five with anti-IL-23. Of the five patients with chronic obstructive pulmonary disease, one received anti-TNF-α therapy, and four received anti-IL-23. Among the four patients with psoriatic arthritis, two received anti-TNF-α, and two received anti-IL-23 therapy. Of the four patients with diabetes mellitus, one was treated with anti-TNF-α, and three with anti-IL-23. The single patient with associated vitiligo received anti-IL-17 therapy.

The PASI and DLQI scores showed a statistically significant decrease between initial moment and after three months of biological treatment. We analyzed the changes in the cytokine levels of TNF-α, IL-17A, IL-17F, and IL-12/23 by comparing their concentrations at 12 weeks to the initial serum levels. The results are summarized in Table 2.

Correlations between PASI and proinflammatory markers can be found in Table 3. A statistically significant moderate positive correlation was found between the PASI score at 12 weeks and the TNF-α serum level at 12 weeks (r = 0.391, *p* = 0.018). A scatter plot with linear regression was used to assess the relationship between PASI score ant TNF-α serum levels after 12 weeks of treatment. The regression line followed the equation y = 2.22 + 0.24x (Figure 1), with an R^2^ of 0.047, indicating that only 4.7% of the variance in TNF-α levels can be explained by PASI score (weak explanatory power). A correlation between PASI either at the beginning of treatment or after 12 weeks and IL-17A, IL-17F, and IL-12/23 serum levels could not be identified.

Lastly, we wanted to see if the treatment class, disease duration, obesity, alcohol consumption, smoking habit, or gender has an effect on serum levels of TNF-α, IL-17A, IL-17F, and IL-12/23 at the onset of treatment or after 12 weeks (Table 4, Table 5, Table 6 and Table 7). We did not find significant differences.

## 4. Discussion

Several studies investigated the serum levels of the main cytokines incriminated in the pathogenesis of psoriasis, mainly by comparing patients with psoriasis with healthy controls or other disease groups. These studies contributed substantially to unravelling the inflammatory mechanisms behind the disease and have stressed out differences in cytokine expression between affected individuals and the general population. However, most of these works are cross-sectional comparisons rather than an evaluation of intra-individual variations in cytokine levels over time. In contrast, our study aims to assess the longitudinal changes in interleukin levels within the same patients following biologic treatment, providing insights into the dynamic immunological response and its potential implications for disease monitoring and personalized therapy.

A meta-analysis, including 57 studies conducted on 2838 psoriasis patients, concluded that increased serum levels of IL-2, IL-17, IL-18, and IFN-γ significantly correlated with psoriasis, highlighting their potential role as biomarkers for monitoring disease activity. These cytokines exhibited significantly higher levels in psoriasis patients compared to controls [4].

Regarding studies examining interleukin levels in psoriasis and their correlation with PASI scores, Wang et al. investigated the relationship between cytokine profiles and psoriasis severity by comparing serum cytokine levels between psoriasis patients and healthy controls. Results showed that IFN-γ, TNF-α, IL-1β, IL-6, IL-17A, IL-18, and IL-23 in psoriasis patients had significantly higher levels compared to controls and a strong correlation between PASI scores and these cytokines [5]. In the study conducted by Fotiadou et al., the serum levels of IL-6, IL-8, IL-17A, IL-22, IL-23, and TNF-α were significantly elevated in psoriatic patients compared to healthy controls. Moreover, patients with active disease exhibited significantly higher levels of IL-17A, IL-23, and IL-22 than those with stable psoriasis, suggesting that these cytokines might play an important role in disease exacerbation [6].

IL-6, IL-20, and IL-22 serum levels were significantly elevated in psoriasis patients compared to controls in the study conducted by Michalak-Stoma et al. Furthermore, a positive correlation between PASI score and IL-22 levels was identified, underlining a link between the inflammation orchestrated by IL-22 and clinical disease severity and, thus, proposing IL-22 as a suitable biomarker for monitoring disease activity in psoriasis. A correlation between IL-23 and IL-17 levels was also observed [7].

Contrarily, serum levels of TNF-α, IL-12/23p40, and IL-17 did not correlate with PASI score, suggesting that these inflammatory markers may not directly reflect disease severity in untreated psoriasis patients; although, TNF-α was significantly higher in psoriasis patients compared to healthy controls. In the same study, a multivariable linear regression model, incorporating PASI as the dependent variable and demographic and clinical characteristics as independent variables, also showed no statistically significant predictors (tested variables: age, gender, disease duration, family history, TNF-α, IL12/IL-23p40, IL-17) of disease severity [8].

The association between psoriasis, metabolic syndrome, and inflammatory cytokines, particularly IL-17, IL-23, and TNF-α, was highlighted by Pirowska et al. Their study concluded that patients with psoriasis and coexisting metabolic syndrome exhibited significantly elevated levels of IL-17 compared to those without metabolic syndrome, suggesting a link between systemic inflammation and metabolic complications in psoriasis [9]. Another study investigated the role of multiple cytokine analysis as an important step for individual immune profile assessment before treatment selection in male psoriasis patients. They observed statistically significant reduced levels of IL-9 in patients with psoriatic arthritis compared to patients without arthritis. Furthermore, they found negative correlations of IL-9, IL-12, and IL-23 with the age of psoriatic patients, IL-12 and IL-23 with disease duration, and IL-6 and IL-9 with the NAPSI score [10]. In our study, no correlation was found between serum interleukin levels either before or after treatment and factors such as gender, disease duration, obesity (BMI > 30 kg/m^2^), alcohol consumption, or smoking status. This may be due to the small number of patients included.

Regarding research upon cytokine levels over time in the same group of psoriasis patients, Laura Mercurio et al. found that IL-38 (member of IL-1 family of cytokines, antagonist) might be a new biomarker to assess the treatment response of psoriasis patients treated with the anti-IL-17A drug, Secukinumab. They stated that skin and circulating levels of the anti-inflammatory IL-38 are reduced in psoriasis patients and the treatment with Secukinumab causes its upregulation and, subsequently, correlates with therapeutic efficacy [11].

In the Lan-Tu-Ya Wu et al. study, 12 patients with moderate and severe forms of psoriasis were enrolled, treated with Secukinumab, and their serum levels of TNF-α, IL-17A, and IL-23 were determined before and after treatment (moment established as patients achieving a PASI score below 1). Secukinumab treatment significantly reduced the serum levels of these cytokines, and moreover, IL-17A and TNF-α were positively correlated with disease duration and age [12].

In the present study, we set the moment of post-treatment evaluation after 12 weeks. Waiting for a complete or almost complete clinical response (PASI score below 1) could have led to a significant reduction in the IL-17A serum levels as well. Contrarily, we did not obtain a correlation between the serum levels of IL-17A or TNF-α and disease duration (long versus short disease duration). Moreover, studies showed that bioactive IL-17A, rather than total IL-17A level, is a more suitable biomarker for monitoring disease activity in psoriasis patients. In our study, a statistically significant decrease in IL-17A levels was not found, maybe due to the assay used, which measures both bioactive and inactive forms (IL-17A attached to inhibitors, regulatory proteins, or antibodies), leading thus to an overestimation of total IL-17A levels [13].

Additionally, IL-6 and IL-22 levels may serve as reliable biomarkers for monitoring treatment efficacy in psoriasis patients. IL-6 appears to be a useful marker in patients receiving anti-TNF-α or anti-IL-12/23 therapy, while IL-22 may be more relevant for those treated with Adalimumab or Infliximab, particularly in long-term disease management over a 36-month treatment period [14]. Therefore, a significant decrease in interleukin levels in our study may also become evident but only over a longer period.

Morita A et al. found that BD2 (beta-defensin 2) is a stable and easily measurable protein that has the potential to serve as a surrogate biomarker for monitoring responses to IL-17A-targeted therapies in clinical practice in a study conducted on 30 psoriasis patients. The serum levels of BD2 underwent a rapid and significant reduction in patients that switched from Cyclosporin A to Secukinumab treatment. BD2 levels demonstrated a strong correlation with the PASI score, with BD2 reductions occurring even before improvements in PASI scores. In conclusion, BD2 might predict a treatment response even before the usual clinical severity scores will [15]. Contrarily, after Guselkumab withdrawal, increases in serum levels of IL-17A, IL-17F, and IL-22 lagged behind PASI worsening, indicating they are poor predictors of psoriasis flare-ups [16].

Various studies focused on the identification of other potential biomarkers for predicting treatment outcomes. In line with this, PAD4 (peptidyl arginine deaminase 4) may serve as a biomarker for the quantification of treatment response in psoriasis patients undergoing anti-TNF-α biological therapy (with Adalimumab or Infliximab), as significant decrease in PAD4 serum levels along with improvements in clinical severity scores was observed. Moreover, the same study reported a decrease in TNF-α levels regardless of the treatment class administered (anti-IL-17 or anti-TNF-α agents) [17], which is consistent with our results, as we observed a statistically significant decrease in TNF-α levels after 12 weeks, across all treatments.

Biological agents reduce the skin involvement in psoriasis patients through multiple mechanisms, thus explaining the correlation between TNF-α serum levels after 3 months of therapy and the improvement of the PASI scores, as demonstrated in the current study. Although the proportion of explained variance remains low, the association supports the potential role of TNF-α as a marker of disease activity. The available anti-TNF-α drugs interrupt the proinflammatory loop powered by different immune cells and keratinocytes. They act rapidly through the downregulation of Th17 cells activity, leading thus to decreased circulating levels of IL-17 [18]. These agents also reduce the expression of CCL20 chemokine in the skin, lowering the Th17 cells recruitment to the psoriatic plaques. Besides, anti-TNF-α drugs are able to downregulate the expression of certain adhesion molecules (E-selectin, ICAM-1, VCAM-1), resulting in reduced leukocyte infiltration into the dermis [19]. IL-23 is a heterodimeric interleukin consisting of 2 subunits (p40 subunit, common with IL-12 and a p19 subunit), both of which have been shown to be increased in psoriatic skin. After binding to its receptor, IL-23 exerts several actions, stimulates the differentiation of ILCs, activates various subsets of T cells, like CD4+, CD8+, and γδ to deliver the proinflammatory IL-17 cytokine, and triggers the macrophages to produce TNF-α. Therefore, countering these mechanisms using biological treatment directed against this cytokine improves the clinical outcome in patients with psoriasis [20]. Anti IL-17A agents reduce the tissue remodeling process promoted by the activation of matrix metalloproteinases and restore the normal differentiation of keratinocytes [21]. Taken together, these mechanisms act synergistically to normalize keratinocyte differentiation and the influx of proinflammatory cytokine in the skin and explain the clinical improvement in psoriasis patients undergoing biological therapy.

Another study conducted by de Magalhães Alves et al. enrolled 262 patients with moderate to severe psoriasis for 6 years, to search if the Th1/Th17 cytokines could serve as predictors for drug survival in psoriasis treatment. They determined for each patient the serum levels of several cytokines, such as IL-2, IL-4, IL-6, IL-10, IL-17A, and TNF-α, before treatment and concluded that a reduced quality of life (scored through DLQI questionnaire) and increased baseline serum levels of IL-6 were associated with treatment discontinuation. Even though IL-6 is not a central cytokine in psoriasis pathogenesis, it may be a suitable biomarker for assessing treatment outcomes [22].

Our study has some important limitations to be taken into consideration when interpreting our findings. First, the small sample size limited the statistical power, especially for subgroup analyses, and prevented the stratification by individual biologic agents due to insufficient numbers within each treatment group. A priori power calculations and larger cohorts will be mandatory in future studies to validate subgroup differences more vigorously. Additionally, the absence of a healthy control group or untreated psoriasis group limits our capacity to conclude whether observed cytokine changes are purely treatment-induced or partially reproduce oscillations in disease activity. Although our study focused on intra-individual changes over time in a real-world clinical setting, we understand the importance of adding comparator groups and cytokine ranges in future studies to better isolate treatment effects. While the 12-week endpoint was based on conventional clinical protocols and biologic treatment timelines, we highlight that more frequent timepoints (e.g., at 24 and 36 weeks) could expose late or more obvious changes in IL-17A, IL-17F, and IL12/23 levels, which could decode better cytokine dynamics. Moreover, the use of ELISA kits measuring total cytokine levels, and not only the bioactive forms, may have reduced the sensitivity to detect biologically eloquent changes. Future studies should use assays targeting bioactive cytokine forms or utilize multiplex panels to improve detection sensitivity. Lastly, the moderate positive correlation and low R^2^ values observed between TNF-α and PASI scores imply limited applicability of TNF-α as a standalone biomarker and emphasizes the intricate relationship between cytokines and psoriasis. Alternatively, we propose that a multimarker technique, incorporating additionally several other cytokines or molecular markers, such as BD-2 for Secukinumab treatment response prediction, PAD-4 for Adalimumab and Infliximab, and IL-6 for patients receiving anti-IL12/23 drugs, may offer greater precision in predicting therapeutic response.

## 5. Conclusions

The present study revealed that, in this group of patients with moderate-to-severe psoriasis, biological therapy decreases TNF-α serum levels and the clinical severity scores of the PASI and DLQI after 12 weeks of treatment. Treatment class, disease duration, obesity, alcohol consumption, smoking habit, or gender showed no effect on the serum levels of TNF-α, IL-12/23, IL-17A, or IL-17F at the onset or after 12 weeks of biological therapy. The PASI score and TNF-α levels correlated at week 12, suggesting that TNF-α could be a useful biomarker to assess clinical response, across all biological therapies. In a larger group of patients, statistical significance may be more important.

## Figures and Tables

**Figure 1 cimb-47-00368-f001:**
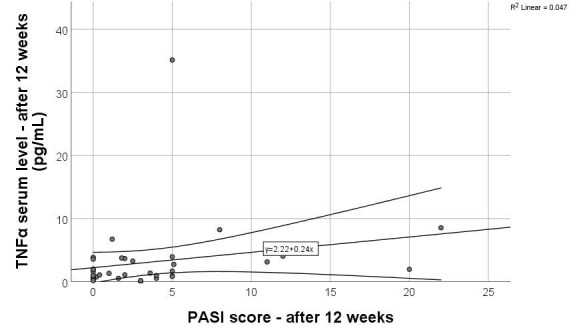
Linear regression analysis between PASI score and serum TNF-α levels after 12 weeks. Dots represent individual PASI score and corresponding serum TNF-α levels after 12 weeks. The solid line indicates linear regression trend, while the dashed lines represent 95% confidence interval of the regression.

**Table 1 cimb-47-00368-t001:** Patient background characteristics.

Patient Characteristics		
Age in years	46 years (30.25; 63.75)	
Gender	Male	20 (55.6%)
	Female	16 (44.4%)
Positive family history	YES	10 (27.8%)
	NO	26 (72.2%)
Disease duration	SDD	9 (25%)
	LDD	27 (75%)
Body mass index	Normal weight	27 (75%)
	Obese	9 (25%)
Alcohol consumption	YES	25 (69.4%)
	NO	11 (30.6%)
Smoking	YES	20 (55.6%)
	NO	16 (44.4)
Pre-biologic oral therapy	YES	36 (100%)
	NO	0
Pre-biologic phototherapy	YES	0
	NO	36 (100%)
Dyslipidemia	YES	10 (27.77%)
	NO	26 (72.22%)
High blood pressure	YES	7 (19.44%)
	NO	29 (80.55%)
Diabetes mellitus	YES	4 (11.11%)
	NO	32 (91.66%)

**Table 2 cimb-47-00368-t002:** Comparisons of severity scores and proinflammatory cytokines between measurements (mean (Q25; Q75)).

	Before Treatment	After 3 Months of Treatment	*p*
PASI	20 (14.25; 23.55)	2 (0;5)	**<0.001**
DLQI	21 (15; 24.75)	4 (0;13)	**<0.001**
TNF-α	2.11 (1.15; 5.45)	1.45 (0.75; 3.63)	**0.012**
IL-17A	4.04 (2.05; 4.80)	2.25 (0.82; 4.26)	0.136
IL-17F	104.68 (33.30; 183.75)	32.65 (19.78; 109.56)	0.162
IL-12/23	218.78 (186.69; 273.96)	215.32 (176.44; 287.21)	0.572

**Table 3 cimb-47-00368-t003:** Correlations between the levels of measured cytokines and disease severity before and after biological treatment.

Variable	PASI Before Treatment		Variable	PASI After 3 Months of Treatment	
	r	*p*		r	*p*
TNF-α before treatment (pg/mL)	0.190	0.266	TNF-α after 3 months of treatment (pg/mL)	**0.391**	**0.018**
IL-17A before treatment (pg/mL)	−0.157	0.359	IL-17A after 3 months of treatment (pg/mL)	−0.014	0.936
IL-17F before treatment (pg/mL)	−0.059	0.789	IL-17F after 3 months of treatment (pg/mL)	−0.090	0.676
IL-12/23 before treatment (pg/mL)	−0.006	0.970	IL-12/23 after 3 months of treatment (pg/mL)	0.002	0.992

**Table 4 cimb-47-00368-t004:** Comparison of the influence of biological agents, disease duration, obesity, alcohol consumption, smoking, and gender on the serum level of TNF-α (mean (Q25; Q75)).

		TNF-α Before Treatment (pg/mL)	TNF-α After 3 Months of Treatment (pg/mL)	*p*
Treatment	Anti IL-17	3.24 (2.14; 6.25)	1.22 (0.75; 1.94)	0.574
	Anti IL-23	1.59 (0.65; 2.81)	0.96 (0.53; 3.12)	
	Anti TNF-α	2.6 (1.32;7.28)	3.66 (1.07; 7.48)	
Disease duration	LDD	2.15 (1.12; 5.57)	1.35 (0.78; 3.56)	0.179
	SDD	1.96 (1.09; 4.75)	1.95 (0.35; 3.79)	
Obesity	0	2.38 (1.32;5.95)	1.55 (0.79; 3.70)	0.579
	1	1.59 (0.46; 2.6)	0.96 (0.53; 3.12)	
Alcohol	0	2.38 (1.67; 5.57)	1.06 (0.53; 3.66)	0.545
	1	1.84 (1.02; 5.33)	1.55 (0.79; 3.50)	
Smoking	0	2.27 (1.04; 5.74)	1.50 (0.53; 3.91)	0.531
	1	1.90 (1.15; 5.45)	1.45 (0.79; 3.22)	
Gender	F	2.22 (0.80; 6.74)	1.93 (0.60; 3.63)	0.427
	M	1.99 (1.15; 4.80)	1.33 (0.75; 3.69)	

**Table 5 cimb-47-00368-t005:** Comparison of the influence of biological agents, disease duration, obesity, alcohol consumption, smoking, and gender on the serum level of IL-17A (mean (Q25; Q75)).

		IL-17A Before Treatment (pg/mL)	IL-17A After 3 Months of Treatment (pg/mL)	*p*
Treatment	Anti IL-17	4.54 (4.19; 6.03)	5.39 (2.34; 7.91)	0.156
	Anti IL-23	2.12 (0.92; 4.51)	1.54 (0.64; 2.93)	
	Anti TNF-α	4.16 (2.47; 5.08)	2.10 (0.87; 3.60)	
Disease duration	LDD	3.37 (1.85; 4.49)	2.02 (0.82; 4.27)	0.608
	SDD	5.37 (3.29; 8.86)	2.56 (1.21; 7.08)	
Obesity	0	4.16 (2.03; 4.79)	2.56 (0.76; 4.36)	0.989
	1	2.91 (1.64; 5.35)	1.77 (1.02; 3.40)	
Alcohol	0	2.89 (2.12; 4.81)	2.58 (0.92; 4.36)	0.639
	1	4.21 (1.62; 5.08)	2.02 (0.74; 4.15)	
Smoking	0	3.80 (2.70; 6.30)	2.25 (1.31; 3.76)	0.115
	1	4.04 (1.08; 4.57)	2.18 (0.73; 5.83)	
Gender	F	3.66 (2.40; 4.50)	2.35 (1.00; 3.99)	0.820
	M	4.19 (1.22;6.36)	2.25 (0.73; 5.83)	

**Table 6 cimb-47-00368-t006:** Comparison of the influence of biological agents, disease duration, obesity, alcohol consumption, smoking, and gender on the serum level of IL-12/23 (mean (Q25; Q75)).

		IL-12/23 Before Treatment (pg/mL)	IL-12/23 After 3 Months of Treatment (pg/mL)	*p*
Treatment	Anti IL-17	214.84 (187.86; 350.05)	227.49 (208.38; 345.07)	0.965
	Anti IL-23	215.86 (158.32; 241.47)	202.96 (160.91; 233.16)	
	Anti TNF-α	226.05 (192.94; 294.63)	230.27 (177.84; 294.28)	
Disease duration	LDD	215.86 (186.66; 290.33)	215.23 (174.09; 272.70)	0.205
	SDD	237.21 (172.64; 266.04)	269.39 (176.63; 321.88)	
Obesity	0	215.86 (176.22; 276.30)	214.51 (183.52; 287.25)	0.949
	1	240.72 (191.11; 278.00)	233.16 (167.05; 304.26)	
Alcohol	0	240.72 (186.76; 339.17)	220.05 (168.32; 287.25)	0.329
	1	215.14 (168.27; 256.39)	214.51 (178.80; 285.35)	
Smoking	0	212.61 (192.37; 296.51)	219.07 (203.73; 283.61)	0.788
	1	224.62 (158.82; 266.49)	201.30 (162.76; 312.84)	
Gender	F	228.29 (178.83; 296.51)	252.17 (172.19; 297.79)	0.738
	M	218.42 (186.84; 246.10)	205.33 (176.44; 232.44)	

**Table 7 cimb-47-00368-t007:** Comparison of the influence of biological agents, disease duration, obesity, alcohol consumption, smoking, and gender on the serum level of IL-17F (mean (Q25; Q75)).

		IL-17F Before Treatment (pg/mL)	IL-17F After 3 Months of Treatment (pg/mL)	*p*
Treatment	Anti IL-17	101.37 (32.78; 224.39)	38.11 (19.02; 328.63)	0.934
	Anti IL-23	105.36 (17.83; 172.31)	23.42 (18.66; 82.56)	
	Anti TNF-α	104.00 (45.69; 187.97)	43.12 (20.00; 170.39)	
Disease duration	LDD	104.68 (29.32; 191.66)	32.65 (17.06; 136.00)	0.612
	SDD	110.45 (42.55; 156.07)	34.24 (22.76; 112.16)	
Obesity	0	99.21 (36.01; 183.75)	35.79 (19.85; 187.04)	0.458
	1	133.43 (23.84; 193.78)	31.11 (18.59; 93.13)	
Alcohol	0	157.00 (71.45; 210.62)	28.45 (18.66; 117.51)	0.210
	1	93.33 (32.93; 108.32)	36.85 (20.77; 110.47)	
Smoking	0	94.42 (34.41; 203.79)	25.37 (19.46; 212.56)	0.705
	1	108.32 (32.63; 172.31)	43.12 (19.32; 108.64)	
Gender	F	157.00 (56.98; 206.04)	28.45 (17.87; 106.81)	0.160
	M	93.33 (26.32; 117.94)	36.85 (21.44; 119.34)	

## Data Availability

The data presented in this study are available on request from the corresponding author due to privacy.

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
