# Peer review of "Assessment of TNF-α, IL-12/23, and IL-17 in Psoriasis: Only TNF-α Reflects Clinical Response After 12 Weeks of Biologic Treatment"

_cimb, 2025, doi:10.3390/cimb47050368_

Round 1

Reviewer 1 Report

Comments and Suggestions for Authors

(1) Please create a patient background TABLE. Also, please add the following patient background: presence or absence of pre-biologic oral therapy (cyclosporine, apremilast, etretinate, etc.), presence or absence of phototherapy, presence or absence of dyslipidemia, presence or absence of hypertension, presence or absence of diabetes mellitus.

(2) Please add a Table of patient background for the TNFα, IL-17A, and IL-12-23 groups and present whether there are any differences in patient background such as PASI score, diabetes, obesity, etc.

(3) Please describe your selection criteria for biologics for your psoriasis patients.

Author Response

1. Please create a patient background TABLE. Also, please add the following patient background: presence or absence of pre-biologic oral therapy (cyclosporine, apremilast, etretinate, etc.), presence or absence of phototherapy, presence or absence of dyslipidemia, presence or absence of hypertension, presence or absence of diabetes mellitus.

Thank you for the helpful suggestion to create a table summarizing the baseline characteristics of the patients. In response to your comment, we have included Table 1 in the Results section, which presents the characteristics as requested.

2. Please add a Table of patient background for the TNFα, IL-17A, and IL-12-23 groups and present whether there are any differences in patient background such as PASI score, diabetes, obesity, etc.

Unfortunately, we were not able to create a separate table summarizing the baseline characteristics and severity scores divided for the anti-TNF-α, anti-IL-17 and anti-IL-23 treatment subgroups due to the limited number of patients with comorbidities in each group. However, the requested information has been included in the manuscript text (Results). We remain open to further suggestions to improve clarity and structure of the data presentation if needed.

3. Please describe your selection criteria for biologics for your psoriasis patients

The selection criteria for biologics in patients is now explained in the 2.1. Design and participants subsection.

Reviewer 2 Report

Comments and Suggestions for Authors

1. The study includes only 36 patients, which is relatively small for subgroup analyses (e.g., treatment types, lifestyle factors). 
2. Only baseline and 12-week follow-up data were collected. 
3. While the study focuses on intra-individual change, including a healthy control group or reference cytokine ranges would provide valuable context for interpreting baseline cytokine levels.
4. Although different biologic classes were used, results were not stratified by individual agents (e.g., Adalimumab vs. Infliximab). Even if sample size is small, this should be discussed as a limitation or noted in future directions.
5. The authors correctly note the limitation of measuring total IL-17A. This important caveat could be discussed earlier in the results section to clarify why IL-17A did not correlate with PASI.
6. Many sentences in the introduction and discussion contain multiple clauses and could be simplified for clarity. For example:
"Biologic therapies targeted against these interleukins have shown significant clinical efficacy..."
could be:
"Biologic therapies targeting these interleukins have demonstrated clinical efficacy..."
7. For the significant correlation between PASI and TNF-α (r = 0.391, p = 0.018), the R² value is only 4.7%, suggesting weak explanatory power. This should be mentioned clearly to temper the interpretation of TNF-α as a biomarker.

Author Response

  1. The study includes only 36 patients, which is relatively small for subgroup analyses (e.g., treatment types, lifestyle factors). 

We acknowledge that the relatively small sample size limits the power for subgroup analyses. This is now explicitly discussed as a limitation in the revised manuscript.

  1. Only baseline and 12-week follow-up data were collected.

We agree that more timepoints would strengthen the findings. However, the 12-week interval was chosen based on standard clinical assessment intervals and biologic treatment response timelines. We have clarified this rationale in the Methods section and highlighted this aspect in the Discussion section of the revised manuscript.

  1. While the study focuses on intra-individual change, including a healthy control group or reference cytokine ranges would provide valuable context for interpreting baseline cytokine levels.

We appreciate this suggestion. While our study was designed to assess intra-individual change rather than between-group differences, we agree that contextualizing cytokine levels is important. We have added a comment on this limitation and potential future inclusion of reference ranges in the Discussion section. However, we did not include a healthy control group in the current study because multiple prior studies—already cited in the Discussion section—have examined cytokine levels in controls compared to psoriasis patients, with results that have been sometimes inconsistent or contradictory, limiting their interpretative value for our specific research objectives.

  1. Although different biologic classes were used, results were not stratified by individual agents (e.g., Adalimumab vs. Infliximab). Even if sample size is small, this should be discussed as a limitation or noted in future directions.

We acknowledge that results were not stratified by individual biologic agents due to limited sample sizes per group. This limitation is now explicitly noted in the revised Discussion section, with a suggestion that future studies with larger cohorts could explore agent-specific effects.

  1. The authors correctly note the limitation of measuring total IL-17A. This important caveat could be discussed earlier in the results section to clarify why IL-17A did not correlate with PASI.

Thank you for highlighting this. We added this aspect also to the Material and Methods section to clearly link the limitation of total IL-17A measurement with its lack of correlation with PASI, and to emphasize the implications for interpretation.

  1. Many sentences in the introduction and discussion contain multiple clauses and could be simplified for clarity. For example:
    "Biologic therapies targeted against these interleukins have shown significant clinical efficacy..."
    could be:
    "Biologic therapies targeting these interleukins have demonstrated clinical efficacy..."

We appreciate this editorial feedback. The manuscript has been reevaluated by a native English speaker. Several modifications have been made to respond to the valuable comments regarding the need for more clarity.

  1. For the significant correlation between PASI and TNF-α (r = 0.391, p = 0.018), the R² value is only 4.7%, suggesting weak explanatory power. This should be mentioned clearly to temper the interpretation of TNF-α as a biomarker.

We agree that an R² value of 4.7% indicates limited explanatory power. This has now been explicitly stated in the Results and further elaborated in the Discussion to temper the interpretation of TNF-α as a biomarker in this context.

Reviewer 3 Report

Comments and Suggestions for Authors

This longitudinal study evaluated serum levels of TNF‑α, IL‑12/23p40, IL‑17A, and IL‑17F in 36 biologic‑naïve patients with moderate‑to‑severe plaque psoriasis before and after 12 weeks of anti‑cytokine therapy, correlating these levels with PASI and DLQI scores. Significant reductions were observed in TNF‑α, PASI, and DLQI, with a moderate positive correlation between week‑12 TNF‑α and PASI. No significant changes or correlations emerged for IL‑12/23p40, IL‑17A, or IL‑17F, nor did patient factors (treatment class, disease duration, obesity, alcohol, smoking, gender) influence cytokine levels.

The ELISA kits quantify total cytokine levels (including inactive/inhibited forms), potentially masking changes in bioactive fractions—consider supplementing with assays specific to bioactive cytokine forms or using multiplex platforms to improve sensitivity.

With only 36 patients subdivided across three biologic classes, the study is underpowered to detect differences for IL‑12/23p40, IL‑17A, and IL‑17F or to robustly assess subgroup effects; a priori power calculation and larger cohort are recommended.

The medians for IL‑17F are reported without interquartile ranges in Table 1, reducing transparency; ensure consistency in data presentation to facilitate interpretation.

The R² of 0.047 in the TNF‑α vs. PASI regression indicates weak explanatory power—this limitation should be quantified and discussed in context of biomarker utility.

The modest correlation and low R² challenge clinical applicability; discuss practicality and compare with other emerging markers (e.g., BD2, PAD4) and consider combining biomarkers to improve predictive value.

A 12‑week endpoint may be insufficient to capture full immunological adaptations; acknowledge that longer-term kinetics (e.g., 24–36 weeks) could reveal significant changes in IL‑17A/IL‑17F and strengthen conclusions.

Without a healthy or untreated psoriasis cohort, it is unclear whether observed serum changes are treatment‑specific or reflect natural disease fluctuation; recommend inclusion of comparators in future studies

Author Response

1- The ELISA kits quantify total cytokine levels (including inactive/inhibited forms), potentially masking changes in bioactive fractions—consider supplementing with assays specific to bioactive cytokine forms or using multiplex platforms to improve sensitivity.

We appreciate this insightful observation. We have added a sentence to the Discussion acknowledging that the use of ELISA kits quantifying total cytokine levels (including inactive forms) may limit sensitivity to detect biologically relevant changes. We also now suggest that future studies consider using assays specific to bioactive cytokine forms or multiplex platforms for improved sensitivity.

2- With only 36 patients subdivided across three biologic classes, the study is underpowered to detect differences for IL‑12/23p40, IL‑17A, and IL‑17F or to robustly assess subgroup effects; a priori power calculation and larger cohort are recommended.

We fully agree that with 36 patients divided across three biologic classes, the study is underpowered for robust subgroup comparisons. This limitation is now explicitly noted in the Discussion section. We have also acknowledged the need for a priori power calculations in future studies with larger cohorts to validate subgroup differences.

3- The medians for IL‑17F are reported without interquartile ranges in Table 1, reducing transparency; ensure consistency in data presentation to facilitate interpretation.

Thank you for pointing this out. We have revised Table 1 (Table 2 in the revised manuscript) to include interquartile ranges for IL-17F to improve transparency and aid interpretation of the data distribution.

4- The R² of 0.047 in the TNF‑α vs. PASI regression indicates weak explanatory power—this limitation should be quantified and discussed in context of biomarker utility.

We agree that this indicates weak explanatory power and now explicitly state this in the Results and Discussion sections. We have added a sentence noting that this finding tempers the interpretation of TNF-α as a biomarker and highlights the complexity of cytokine-disease relationships.

5- The modest correlation and low R² challenge clinical applicability; discuss practicality and compare with other emerging markers (e.g., BD2, PAD4) and consider combining biomarkers to improve predictive value.

We agree that the modest correlation and low R² between TNF-α and PASI suggest limited clinical utility of TNF-α as a standalone biomarker. We also discuss the potential value of a multimarker approach where combining multiple cytokines or molecular markers may enhance sensitivity and specificity in predicting disease activity or therapeutic response.

6- A 12‑week endpoint may be insufficient to capture full immunological adaptations; acknowledge that longer-term kinetics (e.g., 24–36 weeks) could reveal significant changes in IL‑17A/IL‑17F and strengthen conclusions.

We acknowledge that a 12-week endpoint may not fully capture the longer-term immunological changes associated with cytokine-targeted therapies. While our study focused on the initial treatment response window, we have now noted in the Discussion that follow-up beyond 12 weeks (e.g., at 24–36 weeks) may reveal delayed or more robust changes in IL-17A and IL-17F levels, which could offer additional insight into cytokine kinetics and treatment durability.

7- Without a healthy or untreated psoriasis cohort, it is unclear whether observed serum changes are treatment‑specific or reflect natural disease fluctuation; recommend inclusion of comparators in future studies

We agree that the absence of a healthy or untreated psoriasis comparator group limits the ability to determine whether the observed changes in cytokine levels are strictly treatment-induced or partially influenced by natural disease variability. While our study was designed to assess intra-individual change over time in a real-world treatment context, we now explicitly acknowledge this limitation in the Discussion. We also recommend that future studies incorporate healthy controls or untreated psoriasis cohorts to better distinguish between treatment-specific effects and baseline disease fluctuation.

Round 2

Reviewer 1 Report

Comments and Suggestions for Authors

none

Reviewer 2 Report

Comments and Suggestions for Authors

I have no more comments

Reviewer 3 Report

Comments and Suggestions for Authors

Thank you for your revisions